# TASK CALIBRATION FOR DISTRIBUTIONAL UNCERTAINTY IN FEW-SHOT CLASSIFICATION

## ABSTRACT

As numerous meta-learning algorithms improve performance when solving few-shot classification problems for practical applications, accurate prediction of uncertainty, though challenging, has been considered essential. In this study, we contemplate modeling uncertainty in a few-shot classification framework and propose a straightforward method that appropriately predicts task uncertainty. We suppose that the random sampling of tasks can generate those in which it may be hard for the model to infer the queries from the support examples. Specifically, measuring the distributional mismatch between support and query sets via class-wise similarities, we propose novel meta-training that lets the model predict with careful confidence. Moreover, our method is algorithm-agnostic and readily expanded to include a range of meta-learning models. Through extensive experiments including dataset shift, we present that our training strategy helps the model avoid being indiscriminately confident, and thereby, produce calibrated classification results without the loss of accuracy.

## 1 INTRODUCTION

With the great success of deep learning over the last decade, there has been growing interest in investigating methods that are more intuitive to mimic human intelligence. One of the desirable characteristics of human cognition is the ability to learn new information quickly. Few-shot learning is a problem that requires machines to learn with a few examples, and it is usually solved by meta-learning algorithms. Through training over a number of few-shot tasks, the meta-learning model learns to acclimate rapidly to new data and perform desired tasks with the help of prior across-task knowledge. Noteworthy approaches include learning metric space (Koch et al., 2015; Snell et al., 2017; Sung et al., 2018), learning update rule (Ravi & Larochelle, 2017; Andrychowicz et al., 2016), or learning an initialization (Maclaurin et al., 2015; Finn et al., 2017).

Meanwhile, calibration is critical in real-life because the model should correctly inform humans or other models of its degree of uncertainty. Misplaced confidence or overconfidence of a deep network can result in dramatically different outcomes in the decision-making process, such as during autonomous driving (Helldin et al., 2013) or medical diagnoses (Cabitza et al., 2017). Calibration is even more crucial in few-shot learning, for given a few numbers of data at hand, the model is likely to put wrong confidence into the predictions of the unknown data. Also, existing calibration methods, while effective in general classification, are not readily applicable to the few-shot learning. Therefore, we present a novel method to measure task-level uncertainty and exploit it to make a well-calibrated model.

When the model is meta-trained to solve the few-shot classification, it generates a task by randomly choosing classes and sampling support and query examples of corresponding classes. However, we suppose in this way a variety of tasks can be generated, and there are some tasks in which it may be hard for the model to infer the queries from the support examples. Common meta-learning approaches force the model to learn anyway, lacking the discussion about task generation. In this study, we design an algorithm-agnostic calibration method for a few-shot classification model by diminishing the learning signal from those *ill-defined* tasks. This novel training lets the model predict with careful confidence, ultimately obtaining better calibration ability.

To summarize, our contributions are as follows:

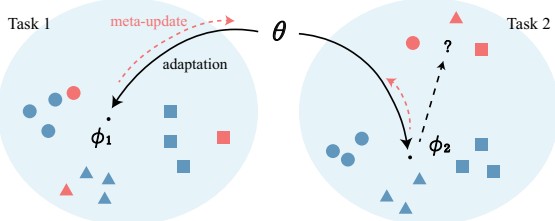

Figure 1: After adaptation to the support examples (blue), inference is made from the query examples (red), which guides the following meta-update. While Task 1 comprises well-distributed $\mathcal{D}_{\text{spt}}$ and $\mathcal{D}_{\text{qry}}$, Task 2 holds a significant support-query discrepancy. We limit the feedback flow from Task 2.

- We propose a simple and straightforward method by which a few-shot classification model learns the ill-defined tasks in a timid way, so as not to place the wrong confidence.
- We extend our approach from an optimization-based to a metric-based framework, confirming that our method is versatile and can be easily expanded to include a range of meta-learning models.
- We demonstrate that we can design an uncertainty-aware model that is suitable for calibration and robustness.

To the best of our knowledge, there has been a work that used weighted meta-update in reinforcement learning literature (Xu et al., 2019) to prioritize and give different weights for each trajectory. However, our work is the first one that quantified task uncertainty and applied it in the few-shot classification framework. Moreover, we validate our work through a dataset shift experiment, which is set to be more challenging and hardly ever addressed due to the degradation of the estimation ability (Snoek et al., 2019).

## 2 PRELIMINARIES

### 2.1 PROBLEM SETUP

A meta-learning problem is composed of tasks from a task distribution $p(\tau)$, and for each meta-training task $\tau \sim p(\tau)$ there is a training set and test set. We denote each as a support set, $\mathcal{D}_{\text{spt}}^{(\tau)}$ and query set, $\mathcal{D}_{\text{qry}}^{(\tau)}$, respectively. In typical few-shot classification scenarios, each task is comprised of data from $N$ randomly selected classes and $k$ support examples per class, with a total of $kN$ examples, $\{(\boldsymbol{x}_s^{(\tau)}, y_s^{(\tau)})\}_{s=1}^{kN} := \mathcal{D}_{\text{spt}}^{(\tau)}$. This is called $N$-way $k$-shot problem. Also, there is a disjoint query set, $\{(\boldsymbol{x}_q^{(\tau)}, y_q^{(\tau)})\}_{q=1}^{k'N} := \mathcal{D}_{\text{qry}}^{(\tau)}$. Meta-training is an iterative process of optimization on $\mathcal{D}_{\text{qry}}$ after being seen $\mathcal{D}_{\text{spt}}$. This setup resembles learning a novel task with only a few examples of training data. The adaptation on $\mathcal{D}_{\text{spt}}$ corresponds to training with the novel task data. Given a few examples, meta-learner is expected to perform predictions on unseen data, which in our setup is $\mathcal{D}_{\text{qry}}$. That is, the meta-learner evolves by acquiring general information across the tasks by optimizing on $\mathcal{D}_{\text{qry}}$.

### 2.2 UNCERTAINTY IN FEW-SHOT CLASSIFICATION

A good meta learner should not only perform well on given tasks but should also make reasonable decisions based on *aleatoric* or *epistemic* uncertainty (Der Kiureghian & Ditlevsen, 2009). While *aleatory* is irreducible uncertainty from data, including class overlap, observation noise, or data ambiguity, *episteme* arises from the stochasticity of the model itself, and can be reduced by obtaining sufficient data. For example, *aleatory* is the case when 'dog' images have inherently high noise (e.g., blurred and corrupted), and thus, have the appearance of other class such as 'cat' (known-unknown). *Episteme* arises when the model is not sufficiently trained for it to match the images to 'dog'.

There is another source, *distributional* uncertainty, which is ascribed to a mismatch between training and test data (Malinin & Gales, 2018). In few-shot classification, this corresponds to the discrepancy between support and query set, often caused by intra-class variations (Chen et al., 2019). For example, *distributional* uncertainty may arise when the 'dog' class includes support examples as 'bulldogs' while by chance, the query is 'chihuahua', resulting in large uncertainty of mapping 'chihuahua' to any class (unknown-unknown). Typical meta-learning algorithms do not consider this mismatch, which we call support-query discrepancy, and naïvely adapt to every observed support

example. Appendix A provides the formalized description of *distributional* uncertainty and its relation to the few-shot learning. In this work, we propose a straightforward method of training an uncertainty-aware model (Figure 1), where we measure the distributional mismatch within a task and constrain the feedback from uncertain tasks.

## 3 TASK CALIBRATION METHOD

Despite the importance of *distributional* uncertainty modeling, the problem setup, which restricts the number of data, renders the model extremely sensitive to the distribution of given data. Therefore, we focus on task-level *distributional* uncertainty instead of considering that of individual data and models. We suggest that this can be implemented by a non-Bayesian approach with an appropriate measure. We propose task calibration model-agnostic meta-learning (TCMAML), a variant of MAML (Finn et al., 2017), that can estimate task *distributional* uncertainty during meta-training and utilize it to modulate the optimization of global parameters. Thereafter, we extend our approach to the metric-based learning model.

### 3.1 BRIEF REVIEW OF MAML

Before introducing our new approach, we provide a brief review of model-agnostic meta-learning (MAML) (Finn et al., 2017) to promote readers' understanding. MAML learns a proper initialization of globally shared $\theta$ that is quickly adjusted to each task $\tau$ by a few gradient descent steps. After being adapted to support examples, MAML updates $\theta$ by optimizing on the query set, minimizing task-specific loss $\mathcal{L}_\tau$ with a task-adapted parameter $\phi_\tau$. This is summarized as the bi-level optimization.

$$\text{inner-level:} \quad \phi_\tau \leftarrow \theta - \beta \nabla_\theta \mathcal{L}(\theta; \mathcal{D}_{\text{spt}}^{(\tau)}) \tag{1}$$

$$\text{outer-level:} \quad \theta \leftarrow \theta - \frac{\alpha}{\mathcal{T}} \sum_{\tau=1}^{\mathcal{T}} \nabla_\theta \mathcal{L}_\tau(\phi_\tau; \mathcal{D}_{\text{qry}}^{(\tau)}) \tag{2}$$

The meta-update (2) aggregates gradients over multiple tasks, so that the tasks are learned simultaneously. If we can measure the task uncertainty, adaptive weighting of task-specific losses is possible, i.e., $\mathcal{L} = \sum_{\tau=1}^{\mathcal{T}} w_\tau \mathcal{L}_\tau(\phi_\tau)$ where $\mathcal{T}$ is the number of tasks in a meta-batch. Each weight reflects the uncertainty of a specific task. If a task $\tau$ has a large uncertainty component, the small value of $w_\tau$ will decrease the contribution of $\nabla_\theta \mathcal{L}_\tau$.

### 3.2 TCMAML

We suggest that class-wise similarity would be a good metric to estimate task uncertainty. As the similarity between classes increases, it becomes harder to distinguish one from another. The similarity is measured in the embedding space mapped by $h_{\phi_\tau}$, the feature extractor before the classifier.

With the pairs of query inputs and labels $(x_q^{(\tau)}, y_q^{(\tau)})$, we collect the average features for every class $c$, $\overline{h}_{\phi_\tau,c} := \frac{1}{k'} \sum_{\{q:y_q^{(\tau)}=c\}} h_{\phi_\tau}(x_q^{(\tau)})$, where $\overline{h}$ maintains feature dimensions. $\overline{h}_{\phi_\tau}$ contains the information about the support and query data because the average features of the query inputs are parameterized by the task parameter $\phi_\tau$, while $\phi_\tau$ is derived from the adaptation on the support data as (1). Therefore, the relations between support and query data determines $\overline{h}_{\phi_\tau}$. We will go deeper in the next section.

We now define the cosine similarity matrix $\mathbf{C}_\tau$, as for all $i, j \in \{1, \cdots, N\}$,

$$[\mathbf{C}_\tau]_{ij} = S_C(\overline{h}_{\phi_\tau,c_i}, \overline{h}_{\phi_\tau,c_j}) := \frac{\overline{h}_{\phi_\tau,c_i}^\top \overline{h}_{\phi_\tau,c_j}}{\|\overline{h}_{\phi_\tau,c_i}\| \cdot \|\overline{h}_{\phi_\tau,c_j}\|} \tag{3}$$

calculates how correlated the two class embeddings are. The diagonal line of $\mathbf{C}_\tau$ is always 1, and the rest of the elements are between $[-1, 1]$. To map the similarity matrix into a scalar, we use Frobenius norm. For $\mathbf{C} \in \mathbb{R}^{N \times N}$, the Frobenius norm of the matrix is $\|\mathbf{C}\|_{\text{F}} = \sqrt{\sum_{i,j} [\mathbf{C}]_{ij}^2}$. From now on, we will simply use a similarity score $s_\tau$ to denote $\|\mathbf{C}_\tau\|_{\text{F}}$.

In few-shot classification, it is important to carefully consider the task generation. More specifically, we prefer to sample the tasks considering how well the task is defined and how much it is reliable to train with. The original meta-loss in (2) assumes equal reliability for uniformly sampled tasks; $p(\tau) = \frac{1}{\mathcal{T}}, \ \forall\tau$. Let $q(\tau)$ be our desired task generating distribution. There is no straightforward way to sample from $q(\tau)$, but we can alternatively use importance sampling.

$$\mathbb{E}_{\tau \sim q(\tau)} \mathcal{L}_\tau(\boldsymbol{\theta}) = \mathbb{E}_{\tau \sim p(\tau)} \frac{q(\tau)}{p(\tau)} \mathcal{L}_\tau(\boldsymbol{\theta}) \approx \frac{1}{\mathcal{T}} \sum_\tau \frac{q(\tau)}{p(\tau)} \mathcal{L}_\tau(\boldsymbol{\theta}) = \sum_\tau q(\tau) \mathcal{L}_\tau(\boldsymbol{\theta}) \qquad (4)$$

Now if we sample the tasks randomly from $p(\tau)$ and compute the importance weight $q(\tau)$, we can easily get the adaptive empirical risk, which allows training with the prior knowledge of task configuration.

TCMAML estimates $q(\tau)$ with a softmax function over $s_\tau$ values. Consequently, we modify the meta-update rule (2) as follows:

$$\boldsymbol{\theta} \leftarrow \boldsymbol{\theta} - \alpha \sum_{\tau=1}^{\mathcal{T}} \nabla_{\boldsymbol{\theta}} \left[ w_\tau \cdot \mathcal{L}_\tau(\boldsymbol{\phi}_\tau; \mathcal{D}_{\mathrm{qry}}^{(\tau)}) \right], \quad w_\tau = \frac{\exp(-s_\tau/T)}{\sum_{\tau=1}^{\mathcal{T}} \exp(-s_\tau/T)} \qquad (5)$$

where $T$ is a scaling factor. Designing the importance weight as above implies large weight on the well-defined tasks, where the measured class-wise similarity is low, and small weight on the ill-defined tasks, where the similarity is high. While the softmax function is an exponential scaling, linear scaling is also possible; $w_\tau = (1/s_\tau)/\sum_{\tau=1}^{\mathcal{T}}(1/s_\tau)$. However, we empirically found that the exponential scaling better performs. Refer to Appendix B.2 for linear scaling TCMAML.

### 3.3 RELATION WITH DISTRIBUTIONAL UNCERTAINTY

The meta-update (5) indicates that for an uncertain task, a small weight value is multiplied, so that its contribution to the updates of $\boldsymbol{\theta}$ is weak. In this sense, TCMAML considers task *distributional* uncertainty by measuring the similarity between the query examples. If we assume $\mathcal{L}$ as a cross-entropy loss, $\boldsymbol{\phi}_\tau$ from task adaptation (1) is an approximation of the maximum likelihood estimate with observed support examples.

$$\boldsymbol{\phi}_\tau = \arg\max_{\boldsymbol{\theta}} \prod_s p(y_s^{(\tau)}|\boldsymbol{x}_s^{(\tau)}, \boldsymbol{\theta}), \quad (\boldsymbol{x}_s^{(\tau)}, y_s^{(\tau)}) \in \mathcal{D}_{\mathrm{spt}}^{(\tau)} \qquad (6)$$

Let $\boldsymbol{\phi}_{\tau^*}$ be the parameter from a task where there is no *distributional* uncertainty. This is satisfied when $\mathcal{D}_{\mathrm{spt}}^{(\tau^*)}$ has exactly the same distribution as $\mathcal{D}_{\mathrm{qry}}^{(\tau^*)}$. We aim to maximize $p(\mathcal{D}_{\mathrm{qry}}^{(\tau^*)}|\boldsymbol{\theta})$ in meta-learning, which is attained when $\boldsymbol{\theta} = \boldsymbol{\phi}_{\tau^*}$, from eq.(6) and $\mathcal{D}_{\mathrm{spt}}^{(\tau^*)} \approx \mathcal{D}_{\mathrm{qry}}^{(\tau^*)}$. We can assume that to make a clear distinction between different classes the embedded representation vectors should be less correlated. Therefore in task $\tau^*$, the query inputs' embedded vectors, $\overline{\boldsymbol{h}}_{\boldsymbol{\phi}_{\tau^*}, c}$, are less correlated because $\boldsymbol{\phi}_{\tau^*}$ optimally distinguishes $\mathcal{D}_{\mathrm{qry}}^{(\tau^*)}$.

In contrast, if there is a significant distributional mismatch between $\mathcal{D}_{\mathrm{spt}}^{(\tau)}$ and $\mathcal{D}_{\mathrm{qry}}^{(\tau)}$, $\boldsymbol{\phi}_\tau$ is the suboptimal solution to maximize $p(\mathcal{D}_{\mathrm{qry}}^{(\tau)}|\boldsymbol{\theta})$, because the parameters cannot infer $\mathcal{D}_{\mathrm{qry}}^{(\tau)}$ from $\mathcal{D}_{\mathrm{spt}}^{(\tau)}$. In the previous example, the parameters adapted from "bulldog" images might not separately distinguish "chihuahua" from another class such

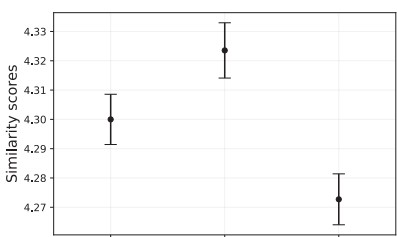

Figure 2: Average similarity scores with 95% confidence interval. *Standard* indicates general few-shot tasks. *Poorly-defined* indicates extremely large support-query discrepancy; images from mismatched classes, and *Well-defined* imitates $\tau^*$. More details in Appendix B.4.

as "cat". This implies higher correlation between the embedded vectors $\overline{\boldsymbol{h}}_{\boldsymbol{\phi}_\tau, c}$. Therefore, we can compare the class-wise cosine similarity, $S_C$, computed using the average features in task $\tau^*$ and $\tau$.

$$S_C(\overline{\boldsymbol{h}}_{\boldsymbol{\phi}_{\tau^*}, c_i}, \overline{\boldsymbol{h}}_{\boldsymbol{\phi}_{\tau^*}, c_j}) \le S_C(\overline{\boldsymbol{h}}_{\boldsymbol{\phi}_\tau, c_i}, \overline{\boldsymbol{h}}_{\boldsymbol{\phi}_\tau, c_j}), \quad \forall\, i, j \qquad (7)$$

From (3) and (5), the above inequality leads to smaller weights on the loss function of uncertain tasks. For a general task $\tau$, a significant mismatch between the distribution of $\mathcal{D}_{\text{spt}}^{(\tau)}$ and $\mathcal{D}_{\text{qry}}^{(\tau)}$ increases the inequality gap.

Figure 2 shows the similarity scores of tasks from different distributions with real-world images. In well-defined tasks as $\tau^*$, the adapted model is good at distinguishing the query representations because they are less correlated. On the other hand, in the extremely discrepant case where the classes of support images do not correspond to those of query images, between-class similarities are high because the model cannot infer the query set from observing the support set. General few-shot classification tasks lie between these well-defined and poorly-defined ones. TCMAML is thus trained by imposing a prior regarding whether a given input task has high *distributional* uncertainty or not.

### 3.4 Extension to the Metric-Based Model

We emphasize that our task calibration approach can be easily extended to a metric-based approach. Unlike optimization-based models such as MAML, a metric-based model does not require a process of explicit adaptation to the task. Instead, the model calculates the distances between the query and support data in a non-parametric way. In this study, we use prototypical network, i.e., ProtoNet (Snell et al., 2017), which is a high-performance metric-based few-shot learning method, and propose and apply our task-calibrated version, TCProtoNet.

Using the distance vector of each query to the support prototypes $\boldsymbol{p} = [p_1, \ldots, p_N]^\top$, we collect the average vectors for each class, $\overline{\boldsymbol{d}}_{\boldsymbol{\phi}_\tau, c} := \frac{1}{k'} \sum_{\{q:y_q^{(\tau)}=c\}} \boldsymbol{d}(\boldsymbol{x}_q^{(\tau)}, \boldsymbol{p})$, where $(\boldsymbol{x}_q^{(\tau)}, y_q^{(\tau)}) \in \mathcal{D}_{\text{qry}}^{(\tau)}$ and $\boldsymbol{d}(\cdot, \boldsymbol{p}) \in \mathbb{R}_+^N$ hold. The assembly of a similarity matrix using the average vectors and the weight computation are effected similarly as in section 3.2. We empirically demonstrate that TCProtoNet performs better calibration than vanilla ProtoNet. The result implies that in a range of few-shot learning models, our method is versatile and applicable to more recent methods such as task dependent adaptive metric, TADAM (Oreshkin et al., 2018).

## 4 Related Work

Uncertainty estimation in deep learning has been persistently studied despite the challenges (Gal, 2016; Lakshminarayanan et al., 2017; Guo et al., 2017; Jiang et al., 2018). Meanwhile, Depeweg et al. (2017) dealt with the decomposition of uncertainty in Bayesian neural networks to better understand and identify their characteristics. Our work also investigates different sources of uncertainty, initially being inspired by Malinin & Gales (2018), which builds Dirichlet Prior Network to explicitly model *distributional* uncertainty. We extend their argument to a few-shot learning framework without requiring any additional parameters or networks. In parallel, Sensoy et al. (2018) parameterized Dirichlet using the evidence vector to regularize predictive distribution, to penalize divergence from the uncertain state.

Few-shot image classification is one of the most challenging tasks for coping with high degrees of uncertainty (Fe-Fei et al., 2003). Estimation and judgment over the uncertainty become challenging when encountered with high-variance data such as a large-scale image set. To overcome this issue, several recent studies have relied on designing Bayesian neural networks. These networks use posterior approximation to quantify parameter uncertainty, which is a type of *epistemic* uncertainty. Probabilistic MAML (Finn et al., 2018) is a variant of MAML (Finn et al., 2017), which approximates the distribution over model parameters by optimizing the variational lower bound. Bayesian MAML (Kim et al., 2018) uses the Stein variational gradient descent for more flexible modeling of posterior. Ravi & Beatson (2019) and Gordon et al. (2019) both amortize approximate inference. Neural Processes (Garnelo et al., 2018) are a combination of a stochastic process and neural network that learns the distribution of functions while representing uncertainty, although they are yet to be proven capable of solving large-scale classification problems. The Bayesian models must compute posterior inference, and most approaches use empirical approximation, which usually requires heavy computations. Conversely, our method uses a non-Bayesian approach, which is computationally advantageous and does not rely on posterior approximation. The TCMAML running time is almost the same as that for vanilla MAML.

## 5 EXPERIMENTS

### 5.1 EXPERIMENTAL DETAILS

We mainly used two different datasets for the experiments: *mini*-ImageNet and CUB-200-2011.

***mini*-ImageNet** (Vinyals et al., 2016) dataset contains 100 classes and 600 images per class. We followed the standard protocol which was initially provided by Ravi & Larochelle (2017), splitting (train, validation, test) set to (64, 16, 20) non-overlapped classes.

**CUB-200-2011** (Wah et al., 2011) dataset is composed of 200 bird species classes incorporating 11,788 fine-grained images. According to Hilliard et al. (2018), it is divided into (100, 50, 50) classes for (train, validation, test) set.

Thanks to their work conducted on fair comparison in the few-shot classification setup, most of our implementation details follow Chen et al. (2019) both in 1-shot and 5-shot tasks. Every accuracy reported is the test accuracy, which was evaluated from 1000 tasks randomly drawn from the test set. By default, we used a Conv-4 backbone, CNN with four convolutional blocks (Vinyals et al., 2016), to not overestimate the performance by using deeper networks. Furthermore, MAML and TCMAML were trained with a first-order approximation of derivatives for the efficiency (Finn et al., 2017).

**Dataset Shift.** Additionally, we conducted *dataset shift* experiments (Chen et al., 2019), in which the train set is the entire *mini*-ImageNet dataset, and validation and test sets are from CUB-200. *mini*-ImageNet $\rightarrow$ CUB-200 was implemented with a ResNet-18 backbone (He et al., 2016) for better convergence. The dataset shift is significantly more challenging than a typical few-shot classification setting because the model has to predict the examples from completely different distributions to those of the train set. This is an intriguing situation from a practical perspective since for most real-world problems it cannot be assumed that new tasks would be from among those with which the model is familiar. However, not much research on uncertainty in few-shot learning has been attempted on shifted datasets.

### 5.2 CALIBRATION RESULTS

Our main contribution is *well-calibrated* classification. The model is well-calibrated when it makes its decisions based on the predicted uncertainty. The correctness of a perfectly calibrated model will match the confidence level (the largest value of the softmax output). For example, the model is *miscalibrated* or overconfident when the confidence is 0.8 on average, but only 70 out of 100 samples are correct. The quantified measures for miscalibration include expected calibration error (ECE) and maximum calibration error (MCE),

$$\text{ECE} = \sum_{m=1}^{M} \frac{|B_m|}{n} |\text{acc}(B_m) - \text{conf}(B_m)|, \quad \text{MCE} = \max_{m \in \{1, \cdots, M\}} |\text{acc}(B_m) - \text{conf}(B_m)| \quad (8)$$

where $\text{acc}(\cdot)$ and $\text{conf}(\cdot)$, respectively, are the empirical accuracy and average confidence for each bin. Here, $B_m$, referred to as bin, is a subset of samples such that the sample's prediction confidence falls into the $m$-th partition. Refer to Guo et al. (2017) for a detailed derivation.

Table 1 and 2 summarize the error rate and accuracy. MAML+TS denotes temperature scaling after training MAML. Temperature scaling (Guo et al., 2017) is one of the calibration methods widely used in general classification, which learns a temperature parameter to control the confidence. It works as post-processing on the validation set. However, in few-shot learning setup, train, validation, and test set all have non-overlapped classes, and the unseen classes are given in the future tasks. Existing calibration methods such as temperature scaling find the best parameter for the known classes, which is not applicable in few-shot classification. This is why MAML+TS produced somewhat random results; a few results are better than MAML but the rest are severely poor.

In most experiments, TCMAML exhibited a significantly decreased error rate than vanilla MAML and ABML, as well as TCProtoNet outperforming ProtoNet. ABML, amortized Bayesian meta-learning (Ravi & Beatson, 2019), is one of the Bayesian methods for few-shot learning and is also based on the MAML algorithm. Notably, while producing calibrated results, TCMAML and TCProtoNet yielded consistent accuracy without incurring the loss of prediction capability. On the other hand, ABML produced poor accuracy with high ECE/MCE rate except for the 1-shot *mini*-ImageNet

Table 1: Calibration results for *mini*-ImageNet experiments. The test accuracy is reported with a 95% confidence interval.

| Methods | 5-Way 1-Shot | | | 5-Way 5-Shot | | |
|---|---|---|---|---|---|---|
| | Accuracy (%) | ECE (%) | MCE (%) | Accuracy | ECE | MCE |
| MAML | $46.92 \pm 0.63$ | 4.04 | 8.77 | $63.18 \pm 0.59$ | 1.46 | 3.98 |
| MAML+TS | $46.92 \pm 0.63$ | 1.39 | 4.59 | $63.18 \pm 0.59$ | 6.01 | 8.79 |
| ABML | $43.47 \pm 0.58$ | 1.48 | 4.82 | $60.06 \pm 0.53$ | 2.97 | 4.93 |
| **TCMAML** | $47.20 \pm 0.63$ | 0.80 | 5.80 | $62.96 \pm 0.58$ | 1.62 | 4.09 |
| ProtoNet | $48.30 \pm 0.62$ | 1.26 | 4.53 | $67.61 \pm 0.54$ | 0.70 | 1.43 |
| **TCProtoNet** | $49.52 \pm 0.63$ | 0.89 | 3.00 | $67.12 \pm 0.53$ | 0.55 | 1.27 |

Table 2: Calibration results (%) for CUB-200 and *mini*-ImageNet $\rightarrow$ CUB-200 experiments. The accuracies are reported in Appendix B.5.

| Methods | CUB-200 | | | | *mini*-ImageNet $\rightarrow$ CUB-200 | | | |
|---|---|---|---|---|---|---|---|---|
| | 5-Way 1-Shot | | 5-Way 5-Shot | | 5-Way 1-Shot | | 5-Way 5-Shot | |
| | ECE | MCE | ECE | MCE | ECE | MCE | ECE | MCE |
| MAML | 2.22 | 2.94 | 2.39 | 3.66 | 18.02 | 35.80 | 18.28 | 25.27 |
| MAML+TS | 8.62 | 13.47 | 2.04 | 3.65 | 16.71 | 35.33 | 20.89 | 27.99 |
| ABML[1] | 3.34 | 5.00 | 2.29 | 4.59 | - | - | - | - |
| **TCMAML** | 1.40 | 2.74 | 2.27 | 3.36 | 15.06 | 27.88 | 17.53 | 24.65 |
| ProtoNet | 3.88 | 6.31 | 0.46 | 4.44 | 4.17 | 6.61 | 0.50 | 1.72 |
| **TCProtoNet** | 2.87 | 5.37 | 0.34 | 0.85 | 3.47 | 4.64 | 0.49 | 1.30 |

[1] ABML is not implemented on dataset shift experiment, due to its inefficiency. Refer to Appendix B.3, regarding the reimplementation of ABML.

case. Also, TCMAML's improvement was more significant in 1-shot experiments compared to the 5-shot, because 1-shot tasks bring more *distributional* uncertainty that induces the wrong confidence of a model. For example, in 5-shot *mini*-ImageNet, MAML was already well-calibrated and there was no more improvement.

Interestingly, TCProtoNet has a better calibration ability than TCMAML. MAML has explicit inner-level optimization, which adapts rapidly to the task, regardless of uncertainty. ProtoNet, on the other hand, updates the parameters using the distance loss between the support prototypes and query set. Therefore, ProtoNet actually warps the embedding space towards reducing the support-query discrepancy. Thus, it can be observed that different learning strategies highly influence the model's calibration abilities.

### 5.3 TASK UNCERTAINTY REFLECTS PREDICTABILITY

One might be curious whether task uncertainty correlates well to the difficulty of a given task. Figure 3 (left) shows that our measure of task uncertainty reflects the model's reliability for making correct predictions. There is a clear trend of low accuracy when the task has large discrepancy in the support and query distribution. In other words, *distributional* uncertainty is strongly related to the classification accuracy. However, the *epistemic* and *aleatoric* uncertainty are much more exacting to measure and do not intuitively explain the difficulty of a task.

We further studied evaluating only those tasks that were estimated to be well-defined. In Figure 3 (right), we excluded the *outlier* tasks of which uncertainty values exceeded the threshold we set. The baseline accuracy for all 1000 tasks is 76.03%. From removing approximately half of all tasks (when threshold is 3.5), the test accuracy reached 81.01% (gaining + 4.98%).

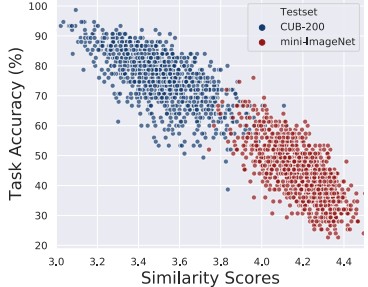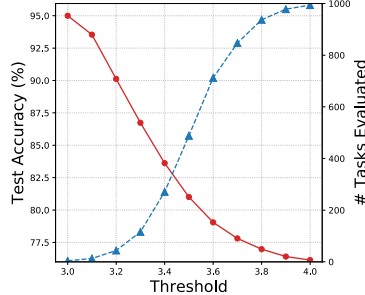

Figure 3: *Left*: TCMAML was trained on 5-shot CUB-200 train set, and thereafter evaluated on different test sets. Each dot represents a test task. 1000 tasks were evaluated for each dataset and each task consisted of 75 query examples, 15 per class to be classified. $y$-axis is task accuracy, the accuracy of 75 queries. *Right*: The test accuracy (solid line) and the number of tasks (dashed line) that were evaluated while excluding the *outlier* tasks. The test set was CUB-200.

## 5.4 META-TRAINING WITH CORRUPTED TASKS

Our task calibration method is robust against data corruption. Meta-training with corrupted data is challenging to the model because it introduces large noise that impedes the accumulation and the transfer of meta-knowledge. Nevertheless, robustness is crucial in that the model should still learn meta-knowledge, even among those tasks, to be transferred to the target tasks.

To this end, we manipulated the task sampling process in meta-train phase. Specifically, with some fixed probability, we corrupted the tasks to include random Gaussian images. That is, a corrupted task consists of 5 support images from *mini*-ImageNet train set and 15 query images that have pixel values from $\mathcal{N}(0, 1)$. We found out that MAML is already robust against this attack due to the informative feature reuse (Raghu et al., 2019) and overfitting to the uncorrupted images; thus, there was no improvement with TCMAML. On the other hand, metric-based model as ProtoNet is fragile because this type of model learns directly from matching the query examples to the support set. If the support and query set are heterogeneous, ProtoNet is easily damaged by unwanted updates.

Table 3: Test results (%) after meta-training under corrupted tasks condition.

| Methods | $p = 0.25$ | | $p = 0.5$ | |
|---|---|---|---|---|
| | Accuracy | Decrease | Accuracy | Decrease |
| MAML | $62.27 \pm 0.57$ | $-0.91$ | $61.12 \pm 0.59$ | $-2.06$ |
| **TCMAML** | $61.82 \pm 0.58$ | $-1.14$ | $61.03 \pm 0.57$ | $-1.93$ |
| ProtoNet | $61.38 \pm 0.55$ | $-6.23$ | $43.60 \pm 0.50$ | $-24.01$ |
| **TCProtoNet** | $63.45 \pm 0.56$ | $-3.67$ | $48.65 \pm 0.53$ | $-18.47$ |

When 25% of tasks were corrupted ($p = 0.25$), ProtoNet performance dropped to 61.38%. Not surprisingly, 50% corruption ($p = 0.5$) led to a further decrease, which is 24.01%p lower than without-corruption training. TCProtoNet managed to exceed its counterpart, 63.45% and 48.65% in 25% and 50% corruption, respectively.

## 6 CONCLUSION

In this study, we asserted that large-scale few-shot classification problems should be examined by carefully estimating uncertainty. We postulate that the few-shot classification contains large discrepancy between the support and query set, although typical meta-learning algorithms do not consider this. We demonstrated that our suggested class-wise similarity score is related to *distributional* uncertainty and reflects the predictability of a task. Further, we demonstrated that task calibration can be extended easily to other methods, such as metric-based learning, via TCProtoNet. The calibration results verified reliable decision-making without the ability to classify being degraded. Lastly,

we found out the task calibration method acquires robustness to the data corruption during meta-training.

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

# A   Detailed Description about Distributional Uncertainty

Here we use the same few-shot classification setup as in the paper. To address the uncertainty in meta-learning, recent works built Bayesian neural networks to formalize posterior distribution over the task-specific parameters $\phi_\tau$, which is an adapted result from $\mathcal{D}_{\text{spt}}^{(\tau)}$ and global parameter $\boldsymbol{\theta}$. The main idea of the Bayesian approach is using the posterior of $\phi_\tau$ as a prior when computing the predictive distribution. Marginalizing over task-specific parameters would give good estimates of an output, considering model's *epistemic* uncertainty.

$$p(y|\boldsymbol{x}, \mathcal{D}_{\text{spt}}, \boldsymbol{\theta}) = \int p(y|\boldsymbol{x}, \boldsymbol{\phi}) p(\boldsymbol{\phi}|\mathcal{D}_{\text{spt}}, \boldsymbol{\theta}) \, d\boldsymbol{\phi} \tag{9}$$

Malinin & Gales (2018) extended above equation to explicitly capture *distributional* uncertainty, via parameterizing a distribution over categorical distribution, $p(\boldsymbol{\mu}|\boldsymbol{x}, \boldsymbol{\phi})$ where $\boldsymbol{\mu}$ stands for the categorical probability; $\boldsymbol{\mu} = [\mu_1, \cdots, \mu_N]^\top = [p(y=1), \cdots, p(y=N)]^\top$. Given enough training data we can assume a point estimate for the model parameters, $\hat{\boldsymbol{\phi}}$. Then (9) can be reformulated as:

$$p(y|\boldsymbol{x}, \mathcal{D}_{\text{spt}}, \boldsymbol{\theta}) = \int \int p(y|\boldsymbol{\mu}) p(\boldsymbol{\mu}|\boldsymbol{x}, \boldsymbol{\phi}) p(\boldsymbol{\phi}|\mathcal{D}_{\text{spt}}, \boldsymbol{\theta}) \, d\boldsymbol{\mu} \, d\boldsymbol{\phi} \tag{10}$$

$$= \int p(y|\boldsymbol{\mu}) p(\boldsymbol{\mu}|\boldsymbol{x}, \hat{\boldsymbol{\phi}}) \, d\boldsymbol{\mu} \tag{11}$$

$$\text{where } \hat{\boldsymbol{\phi}} \sim p(\boldsymbol{\phi}|\mathcal{D}_{\text{spt}}, \boldsymbol{\theta}) = \delta(\boldsymbol{\phi} - \hat{\boldsymbol{\phi}}). \tag{12}$$

$p(\boldsymbol{\mu}|\boldsymbol{x}, \hat{\boldsymbol{\phi}})$ is a prior for categorical distribution, for example Dirichlet prior,

$$p(\boldsymbol{\mu}|\boldsymbol{x}, \hat{\boldsymbol{\phi}}; \boldsymbol{\alpha}) = \frac{\Gamma\left(\sum_{c=1}^N \alpha_c\right)}{\prod_{c=1}^N \Gamma(\alpha_c)} \prod_{c=1}^N \mu_c^{\alpha_c - 1} \tag{13}$$

where $\alpha_c$ is a non-negative concentration parameter for class $c$.

In the case when input has high *aleatoric* uncertainty, $p(\boldsymbol{\mu}|\boldsymbol{x}, \hat{\boldsymbol{\phi}})$ from (13) must be sharp at the center of the $(N-1)$ simplex, being *confident* of imposing uncertain predictions. It is satisfied with symmetric Dirichlet where $\alpha_c$s are the same and larger than 1. Conversely, when there exists high *distributional* uncertainty, desired Dirichlet prior should be *flat*, i.e., $\boldsymbol{\alpha} = [1, \cdots, 1]^\top$. This is because support and query examples form different distributions, so there is no prior belief of mapping query examples to any prediction, $\boldsymbol{x} \mapsto y$.

Although Malinin & Gales (2018) and Sensoy et al. (2018) addressed *distributional* uncertainty by parameterizing Dirichlet prior over categorical distribution, they trained the model with a strong assumption that the distributional mismatch of data is known in advance. However, in few-shot learning problem setup, it is unknown whether the uncertain prediction is attributed to the distribution, model, or data itself. Moreover, every few-shot learning task could potentially involve *distributional* uncertainty due to the limited number of support examples. Thus, it is not easy for the model to decide how much of the meta-learned prior experience it should make use to solve a given task.

We suppose it is more crucial to detect the *distributional* mismatch in few-shot image classification, with an assumption that the *epistemic* uncertainty is relatively lower than the *aleatoric* or *distributional* uncertainty. Since deep networks such as CNN extract highly informative features from the images (Chen et al., 2019; Raghu et al., 2019), uncertainty may not strongly depend on the model's stochasticity. While the Bayesian models can explicitly represent parameter uncertainty in simple problems, it cannot estimate distributional mismatch within complex data. Besides, there is another pitfall. Although few-shot classification constitutes a task with only a small amount of examples, there are in fact large enough data in total and meta-learner is trained under millions of randomly composed tasks. Enough data typically shrink the model posterior and reduce the *episteme*.

# B   Experimental Materials

## B.1   Hyperparameters

We used a Conv-4 backbone except for dataset shift experiments. The Conv-4 architecture (Vinyals et al., 2016) stacks 4 blocks that are each comprised of (Convolution + BatchNorm + ReLU + Max-

Pool). When training, we used data augmentation (RandomResizedCrop, ColorJitter, RandomHorizontalFlip), and each image is resized to the resolution of 84 (224 in the dataset shift). Meta-training lasts 1600 epochs for 5-shot and 2400 epochs for 1-shot experiments, with 25 episodes per epoch. One episode samples 4 tasks ($\mathcal{T} = 4$) and each task samples 15 queries per class ($k' = 15$). We used Adam optimizer (Kingma & Ba, 2014) with learning rate 5e-4. For MAML, the learning rate in the inner-loop update is 1e-2 and the number of inner-loop steps is 5.

In addition, computing weights for each task requires a scaling factor, according to (5). $T$ is a scaling hyperparameter that needs to be tuned, depending on the dataset, type of feature extractor, and the size of a matrix (number of classes), etc. Note that $T \to \infty$ means equally weighted sum, leading to vanilla MAML. In our experiments, we searched $T$ from $\{1/2, 1, 2\}$.

## B.2 LINEAR SCALING TCMAML

In (5), we computed $w_\tau$ with a softmax over the similarity scores. While the softmax function is an exponential scaling (ES), normalizing by linear scaling (LS) is also possible. In this case, we do not need a scaling factor.

$$w_\tau = \frac{1/s_\tau}{\sum_{\tau=1}^{\mathcal{T}} 1/s_\tau} \tag{14}$$

Table 4 exhibits the results of linear scailng TCMAML. However, in most experiments, TCMAML (ES) is comparable with or better than TCMAML (LS). Depending on the similarity score values, linear scaling may have small weight variations across tasks, and produce similar results as MAML.

Table 4: ECE (%) and MCE (%) comparisons between MAML, TCMAML with linear scaling, and TCMAML with exponential scaling.

| Methods | *mini*-ImageNet | | | | CUB-200 | | | | *mini*-ImageNet → CUB-200 | | | |
| | 1-Shot | | 5-Shot | | 1-Shot | | 5-Shot | | 1-Shot | | 5-Shot | |
| | ECE | MCE | ECE | MCE | ECE | MCE | ECE | MCE | ECE | MCE | ECE | MCE |
|---|---|---|---|---|---|---|---|---|---|---|---|---|
| MAML | 4.04 | 8.77 | 1.46 | 3.98 | 2.22 | 2.94 | 2.39 | 3.66 | 18.02 | 35.80 | 18.28 | 25.27 |
| TCMAML (LS) | 1.25 | 4.66 | 1.86 | 4.49 | 2.63 | 3.34 | 1.68 | 2.70 | 16.46 | 39.46 | 18.24 | 24.39 |
| TCMAML (ES) | 0.80 | 5.80 | 1.62 | 4.09 | 1.40 | 2.74 | 2.27 | 3.36 | 15.06 | 27.88 | 17.53 | 24.65 |

## B.3 REIMPLEMENTATION OF ABML

ABML (Ravi & Beatson, 2019) is a Bayesian approach that estimates uncertainty in meta-learning. We reimplemented the ABML algorithm because 1) they only reported *mini*-ImageNet 5-way 1-shot experiment result and 2) our experimental setup is slightly different from theirs. Therefore, we implemented ABML based on our setup.

First, we sampled 4 tasks within an episode as we did in the TCMAML experiments. Second, Ravi & Beatson (2019) did not use data augmentation, but we used the same dataloaders as other experiments of ours. This includes the same data augmentation scheme. Third, we used a first-order approximation of derivatives as TCMAML. For the other setups, such as the number of ensemble networks to train and validate, learning rates, and KL-divergence reweighting factor, we followed the original paper's default values.

However, we did not implement ABML on dataset shift experiment due to its inefficiency of memory and computation. ABML is a Bayesian method, and this implies that ABML is trained by marginalizing over model parameters. In implementation, this requires a significant computation overhead because it should hold several models and receive their gradients in parallel. During the meta-training of ABML, 5 ensemble networks work in parallel, and during the meta-testing, 10 ensemble networks are employed. In Figure 4, TCMAML and ABML show an enormous gap of GPU memory usage, even though they are both MAML-based methods. We can also find that ABML takes a lot of time to compute during the meta-train phase. Because of ABML's inefficiency, we could not apply it to a ResNet-18 model, which is a backbone for dataset shift experiments. It exceeds the capacity of a single GPU we own, and also it takes excessive amount of time to be converged.

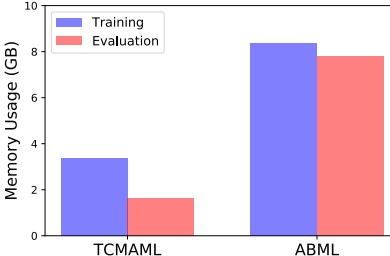 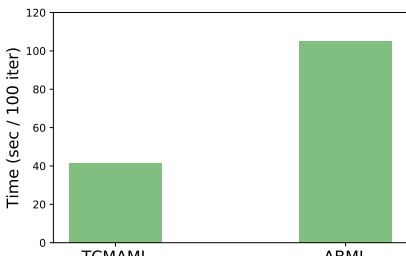

Figure 4: The amount of resource used while training on *mini*-ImageNet 5-way 5-shot with a Conv-4 backbone. Tested on a single GeForce RTX 2080Ti GPU. *Left*: GPU memory usage (GB) during the training (meta-train) phase and the evaluation (meta-test) phase. *Right*: Time consumption (sec) per 100 meta-update iterations in the meta-train phase.

### B.4    EXPERIMENTAL SETUP FOR FIGURE 2

We now describe how we generated Figure 2 in the paper. We measured the class-wise similarity scores of 1000 training tasks, where the tasks are sampled from some manipulated distribution. The dataset we used here is *mini*-ImageNet, and we followed the 5-way 1-shot MAML experiment protocol. We describe the way that we manipulated the task sampling distribution.

**Standard** is a general 1-shot experiment. For the support set, 1 image is sampled from each class. For the query set, 15 images are sampled from the corresponding class.

**Well-defined** tasks are sampled so that there is no *distributional* uncertainty. Although it is hard to perfectly get rid of the uncertainty, it is possible to make the distribution of $\mathcal{D}_{\text{spt}}^{(\tau)}$ and $\mathcal{D}_{\text{qry}}^{(\tau)}$ similar. To this end, we sample 100 support images per class and corresponding 15 query images. By changing few-shot tasks into data-rich *many-shot* tasks, we could simulate on the low uncertainty regime. Adapting to 100 support images (but we did not change the number of adaptation steps) allows $\boldsymbol{\theta}$ to be near optimal to maximize the likelihood of the query set.

**Poorly-defined** tasks are constructed to make extremely large support-query discrepancy, large *distributional* uncertainty. As the **Standard** setting, a **Poorly-defined** task samples 1 support image and 15 query images, but they are from different classes. For instance, the class labels of support images can be {`hair slide`,`carousel`,`wok`,`photocopier`,`jellyfish`}, and the class labels of query images can be {`ant`,`vase`,`dalmatian`,`school bus`,`king crab`}.

### B.5    TEST ACCURACY FOR TABLE 2

In Table 5, we display every test accuracy for CUB-200 and dataset shift experiments. Note that ABML poorly performs in CUB-200 5-shot, while it maintained comparable performance in 1-shot. Also, ProtoNet-based methods outstrip MAML-based methods with large margin in dataset shift experiments.

Table 5: Test accuracy (%) with a 95% confidence interval.

| Methods | CUB-200 | | *mini*-ImageNet → CUB-200 | |
| | 1-Shot | 5-Shot | 1-Shot | 5-Shot |
| --- | --- | --- | --- | --- |
| MAML | $56.76 \pm 0.78$ | $75.62 \pm 0.58$ | $34.24 \pm 0.55$ | $49.34 \pm 0.57$ |
| MAML+TS | $56.76 \pm 0.78$ | $75.62 \pm 0.58$ | $34.24 \pm 0.55$ | $49.34 \pm 0.57$ |
| ABML | $54.99 \pm 0.70$ | $69.59 \pm 0.56$ | - | - |
| **TCMAML** | $56.55 \pm 0.75$ | $76.03 \pm 0.57$ | $33.48 \pm 0.52$ | $50.44 \pm 0.56$ |
| ProtoNet | $57.03 \pm 0.68$ | $77.79 \pm 0.49$ | $44.20 \pm 0.78$ | $64.97 \pm 0.69$ |
| **TCProtoNet** | $56.91 \pm 0.69$ | $77.23 \pm 0.50$ | $45.04 \pm 0.80$ | $64.08 \pm 0.71$ |

# C QUALITATIVE EXAMPLE

| | Task 1 | Task 2 | Task 3 | Task 4 |
|---|---|---|---|---|
| Classes | tank, green mamba, cocktail shaker, bolete, spider web | chime, French bulldog, tobacco shop, jellyfish, clog | snorkel, hotdog, robin, Gordon setter, yawl | fire screen, miniature poodle, tobacco shop, frying pan, ear |

Figure 5: An example of a 4-task batch and their TCMAML weights.

In Figure 5, we depicted an example of weight computation given real images as the input data. This is a sampled batch of tasks, where the minibatch size is 4, from *mini*-ImageNet train set. As it describes, TCMAML gives different weights to each task, regarding the support-query discrepancy. For example, in Task 1, each query image is easily matched to their corresponding support images. The computed weight to this well-defined task is 0.2732. However, Task 4 includes somewhat unclear matching between the support and query images. In this kind of task, even the adapted model cannot confidently distinguish the query images, resulting in the lower weight, 0.1725.

# D RELIABILITY DIAGRAMS

Reliability diagrams (Niculescu-Mizil & Caruana, 2005) visually present model calibration, plotting empirical accuracy versus average prediction confidence, the largest value of the output distribution. Figure 6 describes reliability diagrams comparing between MAML, ABML and TCMAML. We can find that TCMAML has its bars closer to the $y = x$ line than MAML and ABML, implying more reliable predictions. By the way, CUB experiments show slightly different aspects. MAML is *diffident* on its predictions, with the empirical accuracy higher than the confidence. The model which is not sufficiently confident can cause another type of problem. As opposed to the overconfidence, low confidence often occurs when there are ambiguous images with low accuracy. This situation is vulnerable to adversarial attack or perturbation of images (Chakraborty et al., 2018), readily available to decrease overall accuracy due to the model's low confidence. In this case, the model which can accurately tell easy tasks apart is needed. TCMAML again succeeds in calibration with enough confidence in such tasks.

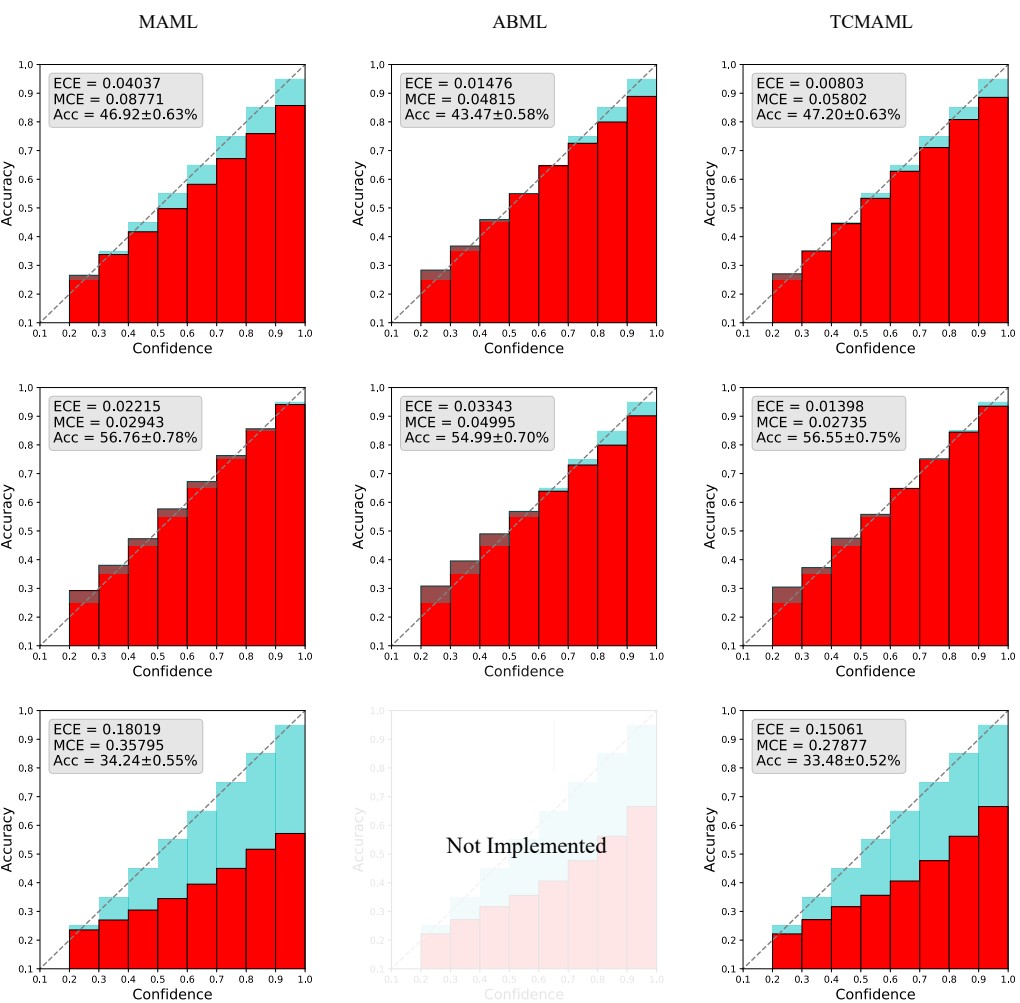

Figure 6: Reliability diagrams from MAML (left column), ABML (middle), and TCMAML (right) classification results. All experiments are 1-shot, and from the top row datasets are *mini*-ImageNet, CUB-200, and *mini*-ImageNet → CUB-200. Red bars indicate the empirical accuracy of the samples whose prediction confidence falls into the specific interval. Perfectly calibrated model should have its bars aligned to $y = x$ line (dashed). Deviation from the line is highlighted as blue if average confidence is higher than accuracy; overconfident, and dark if accuracy is higher; diffident. Note that the confidence is always above 0.2 because there are 5 classes. Best seen in color.

