# OpenReview forum: "Task Calibration for Distributional Uncertainty in Few-Shot Classification"
_ICLR.cc/2021/Conference — Reject_

### Official Review · AnonReviewer1 · 2020-10-25
**Need more to support the contributions**

**Rating:** 5
**Confidence:** 3

**Review:**

This paper presented a task calibration (TC) method, which introduces the notion of "distributional uncertainty", for few-shot classification. Two TC extensions of existing methods (MAML and ProtoNet), namely TC-MAML and TC-ProtoNet, have been presented and experimented. The main contributions of this work the authors pointed out are (1) it's in a non-Bayesian fashion thus is computationally efficient and (2) the TC method can be applied to a range of meta leaning models, and (3) the method is more effective for the few-shot meta-learning situation under dataset shift.

I think this paper deals with an interesting idea but should be improved further. Also the contributions are not clearly supported. The authors should do some more work to demonstrate their contributions. My comments are as below.

1) Regarding the contribution (1), the authors only showed that ABML (amortized Bayesian meta-learning) was computationally expensive, while very efficient Bayesian approximation method based on MC-dropout for the implementation in a Bayesian fashion. The authors should evaluate and compare in terms of the computation time to demonstrate the proposed method is really computationally efficient compared to those in a Bayesian fashion. Also, they need to do something to demonstrate that the proposed method can efficiently handle complex and high-dimensional data sufficiently well.

2) Regarding the contribution (2), there are a variety of meta-learning models that have been presented in recent years (e.g., MetaSGD, TAML, and many more). Which of them can be combined with the TC method? It would be great if the authors investigate the applicability of TC to other existing methods and see if TC improves their performance by experiments as well.

3) Regarding the contribution (3), only one dataset shift case (mini-ImageNet -> CUB-200) was experimented. I strongly recommened the authors to evaulate ther method on other dataset shift cases with additional datasets (e.g., Omniglot).

4) In Table 1 and Table 2, I can't find "significant" improvement of the proposed methods (TC-) compared to their baselines in terms of "classification accuracy" which is the most important metric. In particular in the case of 5-Way 5-shot for mini-ImageNet, TC-MAML was worse than MAML and TCProtoNet was worse than ProtoNet. How, and in which aspect, could you say the proposed methods are "significantly" better?

5) For the organization of this paper, things should be more consistent. For example, accuracy was reported in table 1, but wasn't in table 2 (although provided in appendex B.5). In table 3, the experimental results for MAML and its variants are not provided (althouth the authors mentioned that MAML is already robust for the corrupted tasks).

6) Aren't the methods sensitive to the architectural and training configurations of the CNN models?

7) I think in the current manuscript, there are some strong conjectures that need to be supported by theoretical evidences or more experimental results. For example, "That is, modeling episteme is not much helpful for the calibration in few-shot classification." and the sentences that describe the main contributions.

-----
i've upated my rating after the authors' response.

---

> ### Author Response · Authors · 2020-11-25
> **Thank you for the review and comments.**
>
> Thank you for the comments. We have modified the introduction part to clarify our contributions.
>
> 1. ABML is a Bayesian target calibration method, which is also based on MAML as in our approach. We compared our work with ABML and showed computational inefficiency of ABML in terms of memory and time (see Appendix B.3). We did not consider the MC-dropout since training methods in meta-learning do not use dropout in general.
>
> 2. Our method works by measuring support-query discrepancy. The support-query discrepancy can be computed whenever tasks consist of support and query examples and the backbone contains the features. We thus believe our work is universally applicable to both optimization-based and metric-based approaches. Existing algorithms such as Meta-SGD and TAML are entirely available to work with the task calibration. In this work, we tested the task calibration with MAML and ProtoNet since they are very basic algorithms.
>
> 3. We will include additional dataset shift experiments as you suggested.
>
> 4. The main focus of our experiment is on how the TC method increases the calibration ability in few-shot tasks, not about the classification accuracy. In table 1 and table 2, the important metric is ECE and MCE. Test accuracy is just to validate that TC method does not hurt the classification ability.
>
> 5. We added the MAML result in table 3.
>
> 6. We used Conv4 (miniImageNet and CUB) and ResNet18 (dataset shift) architectures, and the improvements were consistent. Regarding the training configuration, we tried different learning rates and confirmed there was a consistent reduction in ECE and MCE, although we fixed the learning rate for the final results.
>
> 7. We revised some unclear descriptions that is not thoroughly investigated.

---

### Official Review · AnonReviewer3 · 2020-10-26
**I disagree with the paper’s definition of ‘ill-designed’ tasks and their model design based on it.**

**Rating:** 4
**Confidence:** 4

**Review:**

The paper proposes to use class-wise similarity to measure task uncertainty in few-shot learning. In training, if a task has low class-wise similarity, then the task is assigned a higher weight; if a task has high class-wise similarity, then it would be considered ill-defined and assigned a lower weight during training. Experiments on mini-imagenet  and CUB-200 show that it can improve MAML and ProtoNet’s classification performance and uncertainty estimation.

I disagree with the paper’s definition of ‘ill-designed’ tasks. When clases in a training task are very similar, I think it just means that this training task is harder for the model to learn. Therefore, the task at least shouldn’t get a lower weight and somewhat get ignored during training. If at test time, a test task also has very similar classes close to this training task, then this ‘ill-defined’ training task could actually help. With the current method, I think if the data set gets bigger and more complicated, the classification performance will drop significantly. Because the current method would filter out the hard tasks and ignore them during training to some extent.

As for the experiment part, the paper also only conducted experiments on relatively small and simple data sets, mini-ImageNet and CUB-200. If the authors can add an experiment on Tiny-ImageNet and show some performance improvements there, it can make the experiment part much more solid.

The paper says ‘A well-calibrated network often exhibits lower accuracy compared to an overconfident deep network’ in the introduction. It is a false claim. There are a lot of  post-training methods to measure uncertainty at test time. They don’t affect the model’s classification performance since they are post-training methods. Moreover, DeepEnsemble improves the classification performance of a classifier at the same time when it improves the model’s uncertainty estimation.

####################Post Rebuttal#######################################

I meant tiered-ImageNet (https://github.com/renmengye/few-shot-ssl-public). I apologize for the mistake.

I think the rebuttal has not addressed my concerns, especially the one related to the 'ill-defined' tasks. The 'ill-defined' tasks detected from the proposed method could be 'hard but good' tasks benefitting the test. The authors didn't deny this. And the proposed method tries to push the model to learn less from such tasks. A natural deduction is that it could affect the classification performance. I think if the authors can take a further step to disentangle uncertainty estimation and classification, this con of the proposed method can be removed in the future.

---

> ### Author Response · Authors · 2020-11-25
> **Thank you for the review and comments.**
>
> 1. As you mentioned, some tasks might help and should not be ignored during training. Our method works as regularization; it does not get rid of such tasks, but it learns them with small confidence.
> You thought that the definition of ‘ill-defined’ tasks is odd because at test time they might be actually good tasks. Our motivation is that when the meta-learner is trained, there are some tasks which may be hard to infer the queries from the supports, and therefore, accepting the learning signal from those tasks should be diminished. In meta-learning, we actually do not know which tasks will be provided in the future test time. Thus, we try to make an uncertainty-aware model while training.
>
> 2. Tiny-ImageNet contains 200 classes with 500 images for each class, and mini-ImageNet contains 100 classes with 600 images for each class. We are not quite sure mini-ImageNet is a much simpler and smaller dataset than Tiny-ImageNet.
>
> 3. We revised some false claims. Thank you for letting us know.

---

### Official Review · AnonReviewer4 · 2020-10-27
**novelty not clear, tends to cover too many problems but loses focus**

**Rating:** 4
**Confidence:** 3

**Review:**

Overall, this paper studies uncertainty in few-shot classification. The main challenge of this problem is unknown. The description of the problems in the first page is kind of misleading as authors describe many problems, not sure whether authors intend to resolve them all or just part of them. The main contribution of this work is not well motivated.

My main concern is that the contribution of this work is not strong. The main contribution of this work is imposing the class-wise similarity on the meta-update rule in Equation (4). The meta-update is part of the model-agnostic meta-learning pipeline as shown in Equation (2) [Finn et al., 2017]. Equation (4) seems reasonable. However, the novelty of this work is not clear. For example, is the class similarity computation first proposed in this work? Is the weighted meta-update never used in existing literature?

My another concern is that the motivation is not clear. The main problem/problems authors aim to solve are not specified or emphasized. There are many challenges mentioned in this work, such as aleatoric and epistemic uncertainty, few-shot classification, distributional mismatch between support and query data, the trade-off between calibration and the performance of a model. It is not clear what is the main problem that authors aim to solve and how it promotes the contributions of this work.

Figure 1 requires narratives, especially the description about the dash line and solid line with different colors is not clear.

---

> ### Author Response · Authors · 2020-11-25
> **Thank you for the review and comments.**
>
> We have modified the introduction part to more clarify our contributions in this paper, so please check.
>
> 1. To the best of our knowledge, there has been a work that used weighted meta-update in reinforcement learning literature [1] to prioritize and give different weights for each trajectory. However, our work is the first one that quantified task uncertainty and applied it in the few-shot classification.
>
> 2. Our main problem in this work is calibration in few-shot classification. Our motivation is that when the meta-learner is trained to solve the few-shot classification tasks, there are some tasks which may be hard to infer the query data from the support data. Therefore, accepting the learning signal from those tasks should be diminished and this makes the model predict with careful confidence, obtaining the increased calibration ability.
>
> [1] Xu, Zhixiong, Lei Cao, and Xiliang Chen. "Meta-learning via weighted gradient update." IEEE Access 7 (2019): 110846-110855.

---

### Official Review · AnonReviewer2 · 2020-10-28
**task calibration by class-wise similarity across tasks**

**Rating:** 5
**Confidence:** 4

**Review:**

The paper presents a task calibration method to for meta learning, aiming at better task uncertainty estimation. The method modifies the MAML meta-learning approach by weighting the task-specific loss using class-wise similarity, measured by the cosine similarity of the task features. The authors further propose to use this weighting on metric-based models for few-shot classification.

The change in weighting proposed by the paper, although incremental, provides an appreciable performance improvement of existing meta-learning methods. However, some comparisons to existing uncertainty calibration methods are recommended to show a more comprehensive evaluation. In particular, one claim is its advantages over Bayesian methods. Probabilistic MAML [1] and Bayesian MAML [2] could be readily applied as additional comparisons to the author’s proposed method. For metric-based methods, as the authors indicated, it may worth testing with more recent developments, including TADAM.

An additional experiment could also be useful to verify that the class similarity weighting is performing its intended purpose. For example, one can construct a task where the labels are shuffled as in Zhang et al. [3]. Because the labels are uninformative, the task should not be useful for meta learning and should provide almost zero weight.

The description of task corruption experiments can be improved. Were the reported results based on the total four tasks for meta-training only as described in Appendix B.1? Also, it may worth putting the results from MAML variants in addition to ProtoNet.

The authors motivated the method with the discussion of distributional uncertainty. A more explicit characterization relating the posterior predictive distribution to the proposed method would strengthen the paper.

There are also several typos in the current submission. The presentation can be improved for better readability in section 3.2. For example subscripts for classes in equation 3 can be changed to be different from the ones for the query inputs and labels to avoid confusion.



[1] Finn et al., Probabilistic Model-Agnostic Meta-Learning.
[2] Yoon et al., Bayesian Model-Agnostic Meta-Learning, NuerIPS 2018.
[3] Zhang et al., Understanding deep learning requires rethinking generalization.

----post rebuttal----

I am downgrading my score as the authors did not address most of my concerns. The paper can be further improved with the suggested experiments as well as discussion on distributional uncertainty but the revised version does not appear to contain any of these suggested changes.

---

> ### Author Response · Authors · 2020-11-25
> **Thank you for the review and comments.**
>
> We revised our paper based on your comments, and modified the introduction part to clarify our contributions. Also, we will add some experiments as you mentioned, including BMAML or TADAM.
>
> If we use a task with random labeling, we are completely sure that this task will get almost zero weight. Although we did not exhibit a random labeling experiment, you can find a qualitative example of real task images and weight distribution in our revised version, Appendix C.
>
> Task corruption experiments were conducted with the same training configurations as described in Appendix B.1.

---

### Decision · Program_Chairs · 2021-01-07
**Final Decision**

**Decision:**

Reject

**Comment:**

This paper tries to address the uncertainty calibration problem in meta-learning by weighting the gradient from different tasks according to class-wise similarity. There have been many concerns raised by the reviewers and most of them either are still not properly addressed after the rebuttal period.

The main concerns are as follows:
- The problem the paper tries to address is not clear. The use of weighting in meta-update is motivated from distributional uncertainty, but it is not clear how that will improve the task calibration at meta-testing time.
- The proposed update runs into the risk of focusing on simple tasks and down-weighting hard tasks that could be improved with more learning to get more discriminative features. That might hurt the classification performance even though it gets better calibration quality.
- Novelty is limited. The proposed method is a fairly small modification on the original MAML algorithm.
- More comprehensive empirical evaluation is required to support the superiority of the proposed method to other baselines.

I suggest the authors take all the reviewers' comments seriously and improve their work for a better revision.